# Single sperm karyotyping of testicular sperm in non-obstructive and obstructive azoospermia using next generation sequencing

Sumiko Sueyoshi[1,2]☉, Akifumi Ijuin[1]☉, Hiroe Ueno[1]☉, Ai Miyakoshi[1], Haru Hamada[1], Misaki Toda[1], Naoki Tsuchiya[1,2], Miki Tanoshima[3], Mayuko Kurumizaka[1], Marina Saito[1], Yuki Oike[1], Kazumi Takeshima[1], Teppei Takeshima[1], Shinnosuke Kuroda[1], Yasushi Yumura[1], Shin Saito[4], Ryoko Asano[4], Taichi Mizushima[2], Etsuko Miyagi[2], Hideya Sakakibara[4], Hiroki Kurahashi[5], Akira Yanagihara[5,6], Mariko Murase[1], Tomonari Hayama[1,4]*

1 Center for Reproductive Medicine, Yokohama City University Medical Center, Yokohama, Kanagawa, Japan, 2 Department of Obstetrics and Gynecology, Yokohama City University School of Medicine Graduate School of Medicine, Yokohama, Kanagawa, Japan, 3 Department of Genetic Diagnosis and Therapy, Yokohama City University Medical Center, Yokohama, Kanagawa, Japan, 4 Department of Gynecology, Yokohama City University Medical Center, Yokohama, Kanagawa, Japan, 5 Division of Molecular Genetics, Institute for Comprehensive Medical Science, Fujita Health University, Toyoake, Aichi, Japan, 6 Research and Development Department, OVUS CO., LTD, Toyoake, Aichi, Japan

☉ These three authors contributed equally to this work.
* tommy_h@yokohama-cu.ac.jp

## Abstract

The sperm of infertile men have higher rates of chromosomal abnormalities than those of fertile men. Miscarriage rate is also higher following testicular sperm extraction combined with intracytoplasmic sperm injection (TESE-ICSI). Sperm chromosomal abnormalities are assumed to be the cause of miscarriages. Previous testicular sperm karyotyping studies have only examined a few selected chromosomes using fluorescence in situ hybridization. The aim of this study was to provide a more detailed analysis of sperm karyotyping by analyzing all chromosomes using next-generation sequencing (NGS) in clinically usable testicular sperm. Sperm discarded after clinical use was collected for NGS. Additionally, sperm were individually collected by micromanipulation from patients with obstructive azoospermia (OA) and non-obstructive azoospermia (NOA) who underwent TESE-ICSI. For comparison, ejaculated sperm from control and balanced translocation (BT) carriers were examined. Karyotyping was performed on individual sperm cells using NGS. The number of normal and aberrant sperm was compared. Seventeen patients participated in this study: control (n=4), BT (n=3), OA (n=5), and NOA (n=5). Ten sperm samples per patient were analyzed. The total acquisition rate for single sperm karyotyping was 85% (145/170). Karyotyping of sperm from the BT group revealed sperm with unbalanced chromosomes derived from carrier translocations. Among the NOA group, 7/41 (17%) sperm samples exhibited aberrant karyotypes, whereas no aberrant sperm

**Data availability statement:** All relevant data are within the manuscript and its Supporting information files.

**Funding:** The author(s) received no specific funding for this work.

**Competing interests:** The authors have declared that no competing interests exist.

were identified in the control and OA groups. Individual differences were observed in the frequency of sperm chromosomal abnormalities among patients with NOA. In conclusion, sperm chromosomal abnormalities are frequently observed in patients with NOA even after sperm selection for clinical use. As the frequency of chromosomal abnormalities varies among patients with NOA, single sperm sequencing may help identify patients with NOA most likely to benefit from PGT-A.

## Introduction

Although miscarriage is considered a rare event, 15.3% of clinically recognized pregnancies result in miscarriage [1], and women who undergo miscarriage experience mental and physical distress. Most miscarriages are caused by chromosomal abnormalities in the embryos. Examination of trophoblastic specimens from spontaneous abortions indicates that 60% of specimens had chromosomal abnormalities [2]. More than half of the embryonic chromosomal abnormalities are caused by chromosomal abnormalities in oocytes [3], and the miscarriage rate increases with increasing female age [4]. However, 5–10% of chromosomal abnormalities in embryos are of male origin [5]. Hence, sperm are also responsible for chromosomal abnormalities in embryos.

The rate of chromosomal abnormalities is 4% in sperm from fertile men with normal semen parameters [6] and 8.4% in infertile patients with normal semen parameters [7]. This rate increases to 13% in sperm from patients with oligoteratoasthenozoospermia (OTA), defined as abnormal sperm morphology under insufficient motile sperm count, and 17.6% in sperm collected by testicular sperm extraction (TESE) from patients with azoospermia [8]. This indicates that chromosomal abnormalities occur more frequently in sperm from infertile men than in sperm from the general population, and the severity of infertility is proportional to the frequency of sperm chromosomal abnormality [9,10]. Furthermore, TESE combined with intracytoplasmic sperm injection (TESE-ICSI) in azoospermia couples shows a higher miscarriage rate than that of the general population [11]. Bettio et al. analyzed the conceptuses of miscarriages following TESE-ICSI. They reported that 80% of conceptuses had chromosomal abnormalities [12]. When preimplantation genetic testing for aneuploidy (PGT-A), which examines embryonic chromosomes through blastocyst biopsy, is performed, the proportion of monosomic and trisomic embryos is higher in the TESE-ICSI embryos than in normo-spermic embryos [13]. Severe male infertility is an indication for PGT-A because sperm chromosomal abnormalities are considered to cause high aneuploidy rates in embryos as well as increased high miscarriage rates.

The chromosomal analysis of sperm is more complicated than that of somatic cells. The DNA molecules in somatic cell nuclei are coiled and aggregated around histones, whereas the DNA in sperm nuclei is tightly wrapped around protamine and highly condensed, making DNA extraction and staining difficult. In 1990, Wyrobek et al. established a fluorescence in situ hybridization (FISH) assay for detecting the Y chromosome [14]. Subsequently, karyotype analysis of sperm using FISH became

the standard method. FISH is widely used for its convenience and low cost, however, conventional FISH methods are limited to examining only a few selected chromosomes. Because FISH uses fluorescently labeled DNA probes against specific chromosomal sites to detect aneuploidy, structural chromosomal abnormalities of individual sperm cannot be detected. To examine all chromosomes, sperm karyotyping analysis using automated whole-chromosome FISH [15], array comparative genomic hybridization [16,17] and next-generation sequencing (NGS) [18] has recently been proposed.

Sperm karyotyping studies are shifting toward karyotyping of all chromosomes. These studies are conducted on men with normal semen parameters because analyzing all sperm chromosomes is technically difficult and it requires a large amount of sperm. Testicular sperm obtained using TESE, which are few and difficult to isolate, have only been karyotyped using conventional FISH methods [8,19,20]. To date, no studies have reported performing karyotyping of testicular sperm encompassing all chromosomes. Therefore, analyzing all chromosomes in testicular sperm is essential.

Azoospermia is classified into non-obstructive azoospermia (NOA) and obstructive azoospermia (OA). The fertilization and pregnancy rates of TESE-ICSI in patients with NOA are significantly lower than those in patients with OA [21]. NOA, which is caused by spermatogenic failure, is a more severe form of infertility than OA, which is caused by sperm duct obstruction. As mentioned above, the proportion of sperm with chromosomal abnormalities increases in proportion to the severity of infertility. NOA is expected to have higher rates of chromosomal abnormalities in the sperm than OA. Therefore, we hypothesized that fertilization and pregnancy rates are lower in cases of NOA than in those of OA because of the higher proportion of sperm with chromosomal abnormalities in NOA. However, previous sperm karyotyping studies have only analyzed testicular sperm from patients with NOA [8,19,20]. To investigate whether a difference in the rate of sperm with chromosomal abnormalities exists between patients with NOA and patients with OA, analyzing testicular sperm from both patients with NOA and patients with OA is necessary.

Sperm mature and gain motility as they pass through the epididymis, therefore, most testicular sperm are immature, immotile and have poor morphology. Consequently, the total motile sperm count recovered by TESE in patients with azoospermia is markedly low, ranging from several hundred to several thousand [22]. In addition, frozen testicular sperm is often thawed and used for TESE-ICSI, and the stress of freezing and thawing further reduces the number of surviving sperm [23]. Although only a few to several dozen clinically usable sperm can be obtained from a single thaw, mature sperm without morphological abnormalities have to be selected for ICSI through microscopic observation. Previous studies on karyotyping of testicular sperm using the conventional FISH method have reported analyzing all sperm regardless of morphology and developmental stage [8,19,20]. Karyotyping in these studies was performed even on sperm with poor morphology and on immature sperm cells. Thus, the results obtained in these studies may not reflect the chromosomal conditions of clinically usable sperm. Therefore, it is necessary to select and analyze mature testicular sperm with good morphology to reflect the karyotype of clinically usable sperm.

In this study, we karyotyped testicular sperm using NGS to obtain karyotyping analysis of all chromosomes. We isolated mature testicular sperm with good morphology using micromanipulation to obtain results that reflected the karyotype of the sperm used for injection into oocytes. We examined sperm from the normal control, OA, and NOA groups and analyzed their sperm karyotypes using NGS to compare the number of sperm with chromosomal abnormalities to those with normal chromosomes. A balanced translocation (BT) carrier group was included as the quality control for karyotyping. We analyzed sperm from patients with both OA and NOA to examine whether there was a difference in the rates of sperm with chromosomal abnormalities between patient with OA and NOA.

## Materials and methods

### Ethical approval

This study was approved by the Ethics Committee of Yokohama City University (approval number: A171130002). Written informed consent was obtained from all the patients, and the study was conducted in accordance with the principles of the Declaration of Helsinki and the Ethical Guidelines for Medical and Biological Research Involving Human Subjects.

## Patient recruitment

This study was conducted between October 1st, 2021 and July 30, 2024 at the Center for Reproductive Medicine at Yokohama City University Medical Center. For the control group, four patients with normozoospermia who underwent IVF for female infertility were recruited. All men in the control group met the criterion of having a total motile sperm count of $15 \times 10^6$/mL or more in semen tests. In the BT group, three BT carriers who underwent IVF for PGT were recruited. For the OA group, five patients with OA who underwent TESE for azoospermia were recruited. For the NOA group, five patients with NOA who underwent TESE for azoospermia were recruited (Fig 1).

As there have been few reports on sperm NGS, we needed to ensure that analysis was performed correctly. For quality control purposes, we analyzed sperm from BT carriers whose karyotype data had already been obtained from the peripheral blood. BT carriers have a normal gene dosage and phenotype; however, during gametogenesis, unbalanced translocated chromosomes are distributed to the gametes (Fig 2). We hypothesized that if we could detect sperm with unbalanced chromosomes derived from the patient's translocation, it would confirm the accuracy of our karyotyping protocol.

## Sperm preparation

For the control and BT groups, patients were instructed to 2–7 days sexual abstinence before semen sample collection. A total motile sperm count $>15 \times 10^6$/mL was selected as the cut-off value for the control group. Ejaculated semen was liquefied and examined to evaluate the volume, sperm concentration, and total sperm motility rate, according to the WHO laboratory manual for the examination and processing of human semen, sixth edition [24]. Liquefied semen was gently layered on top of a density gradient centrifugation medium (0.3 mL of low-density solution consisting of 45% sperm separation medium (SpermGrad™, Vitrolife, Gothenburg, Sweden) and 55% of the culture medium (SpermRinse™, Vitrolife, Gothenburg, Sweden) on top of 0.3 mL high-density solution (90% sperm separation medium and 10% culture medium) and centrifuged at $500 \times g$ for 20 min. After the supernatant was removed,

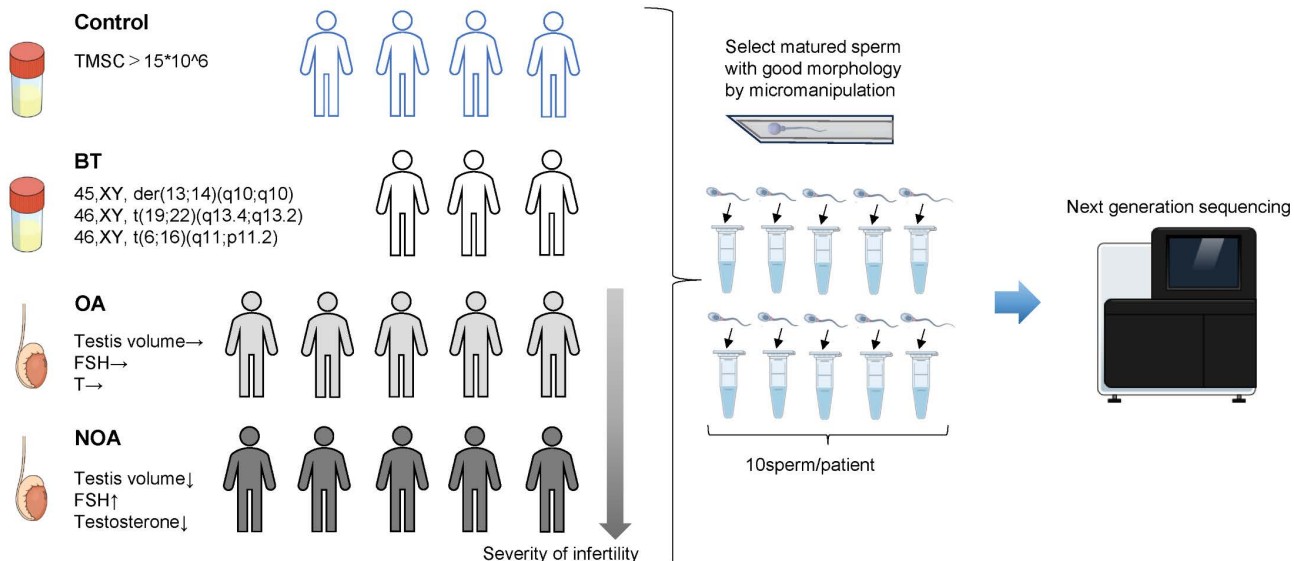

**Fig 1. Experimental overview.** Seventeen patients participated in this study: control group (n = 4), BT group (n = 3), OA group (n = 5), and NOA group (n = 5). Ten sperm per patient were analyzed by NGS. BT, balanced translocation carrier; OA, obstructive azoospermia; NOA, non-obstructive azoospermia; TMSC, total motile sperm count; FSH, Follicle-stimulating hormone; T, testosterone.

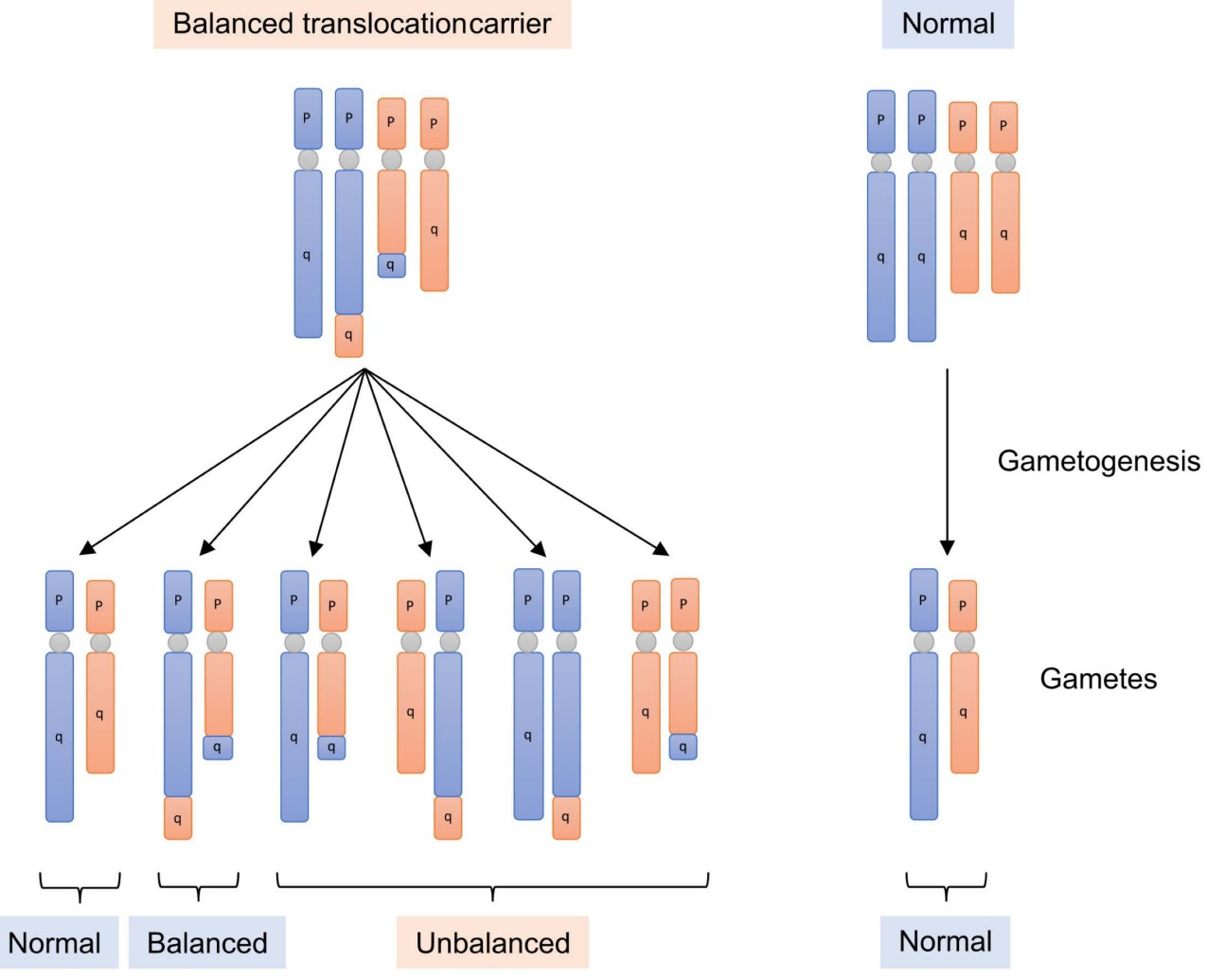

**Fig 2. Gametogenesis of balanced translocation carrier.** Balanced translocation carriers have a normal gene dosage and phenotype; however, during gametogenesis, unbalanced translocated chromosomes are distributed to the gametes. Examples of gametes formed by alternate segregation, adjacent-1 segregation and adjacent-2 segregation are shown.

the precipitate was washed with 5 mL of culture medium and centrifuged at 200 × g for 10 min. The supernatant was then removed, and 1 mL the culture medium was layered on the precipitate and incubated for 20 min at 37°C at an angle of 30°. Next, actively motile sperm with 200 µL of supernatant was removed and placed in a new tube for insemination.

Testicular sperm were cryopreserved after surgical intervention and thawed for insemination. The thawed sperm samples were placed in 2 mL culture medium. The culture medium and sperm samples were pipetted lightly for homogenization, gently layered on top of the density gradient centrifugation medium (0.3 mL of low-density solution [45% sperm separation medium and 55% culture medium] on top of 0.3 mL a high-density solution (90% sperm separation medium and 10% culture medium), and centrifuged at 500 × g for 20 min. After removing the supernatant, the precipitate was washed with 5 mL of culture medium and centrifuged (150 × g, 10 min). The supernatant was removed, and sperm was used for injection into the oocyte.

## Single sperm isolation by micromanipulation

After sperm preparation and clinical use, the discarded sperm were collected for karyotyping. Ejaculated sperm from patients in the control and BT groups were collected individually by micromanipulation using ICSI injection pipettes (inner diameter: 4.5–5.5 µm). Motile sperm with an elongated head and a normal tail were selected. After selection, each sperm was transferred by mouth pipette to individual tubes and frozen at −80°C with 2 µL of PBS/PVP (PGT test medium, OVUS, Nagoya, Japan). For the testicular sperm of patients with OA and NOA, sperm were collected individually by micromanipulation using ICSI injection pipettes (inner diameter: 4.5–6 µm). Sperm with an elongated head and a normal tail, regardless of motility, were selected. After selection, each sperm was transferred to individual tubes and frozen.

## Karyotyping of single sperm with NGS

Single sperm DNA was prepared using the MALBAC method. The sperm pretreatment solution was added to a microtube containing a single sperm, and the samples were heat-treated using a thermal cycler. After heat treatment, a stop solution was added. Whole genome amplification (WGA) was performed using WGA reagents from the Ion ReproSeq PGS kit (Thermo Fisher Scientific, Waltham, MA, USA) according to the manufacturer's protocol. The sequencing library was prepared using a library preparation program for NGS aneuploidy analysis on the Ion Chef Instrument (Thermo Fisher Scientific, Waltham, MA, USA). NGS was performed using the Ion GeneStudio S5 Plus System (Thermo Fisher Scientific, Waltham, MA, USA) with an Ion 530 chip, according to the manufacturer's protocol. The ReproSeq PGS Kit can detect of whole chromosome and chromosome arm copy number events (down to 17 Mbp copy number events). Data analysis was performed using the aneuploidy detection program of Ion Reporter Software (Thermo Fisher Scientific, Waltham, USA). The reference sequence selected in Ion Reporter Software is hg19.

## Statistical analysis

The Student's T- and the Kruskal–Wallis tests were used to compare patient characteristics. The Fisher's exact test was used to compare the success rates of karyotyping between the testicular and ejaculated sperm. Pearson's product-moment correlation coefficient was used to determine whether individual karyotyping success rates were correlated with sperm count at TESE in patients with NOA and OA. Pearson's product-moment correlation coefficient was used to determine whether individual aberrant sperm detection rates were correlated with sperm count at TESE in patients with NOA and OA. Statistical significance was set at $P < 0.05$.

## Results

### Patients' background

The characteristics and histories of the seventeen patients and their sperm parameters are detailed in Tables 1 and 2. The median age of the patients was 43.5 years in the control group, 37.0 years in the BT group, 39.0 years in the OA group, and 35.0 years in the NOA group, with no significant difference across the four groups (Table 2).

Peripheral blood karyotypes in the BT group were 45,XY, der(13;14)(q10;q10) in patient #5, 46,XY, t(19;22)(q13.4;q13.2) in patient #6, 46,XY, t(6;16)(q11;p11.2) in patient #7 (Table 1). Peripheral blood karyotyping was not examined in the control group because it was not required for their treatment. The patients' peripheral blood karyotypes in the OA and NOA groups were 46,XY except for two patients who did not undergo peripheral blood karyotyping for financial reasons.

The mean serum luteinizing hormone (LH) level was 3.7 mIU/mL in the OA group and 11.0 mIU/mL in the NOA group ($P = 0.04$). The mean serum FSH level was 4.6 mIU/mL in the OA group and 22.2 mIU/mL in the NOA group ($P = 0.007$). The mean serum testosterone level was 5.0 ng/dL in the OA group and 3.5 mIU/mL in the NOA group, respectively

**Table 1. Patient history and clinical characteristics.**

| Patient # | Age | Group | Peripheral blood karyotype | AZF deletion | Semen volume (ml) | Sperm concentration (million/ml) | Motility (%) | Total motile sperm count | Serum LH (mIU/mL) | Serum FSH (mIU/mL) | Serum Testosterone (ng/mL) | Testis Volume (R/L ml) |
|---|---|---|---|---|---|---|---|---|---|---|---|---|
| 1 | 38 | Control | – | – | 2.6 | 57.4 | 32.0 | 47.8 | – | – | – | – |
| 2 | 40 | Control | – | – | 2.2 | 16.5 | 42.4 | 15.4 | – | – | – | – |
| 3 | 47 | Control | – | – | 3.1 | 54.0 | 50.9 | 85.2 | – | – | – | – |
| 4 | 47 | Control | – | – | 3.2 | 31.0 | 48.4 | 48.0 | – | – | – | – |
| 5 | 33 | BT | der(13;14) | – | 2.4 | 28.6 | 21.5 | 14.8 | – | – | – | – |
| 6 | 37 | BT | t(19;22) | – | 5.8 | 2.6 | 7.7 | 1.16 | – | – | – | – |
| 7 | 42 | BT | t(6;16) | – | 2.0 | 36.5 | 51.1 | 37.3 | – | – | – | – |
| 8 | 35 | OA | ND | ND | – | – | – | – | 6.9 | 5.2 | 9.38 | 20/16 |
| 9 | 38 | OA | 46,XY | AZFcgr/gr | – | – | – | – | 2.1 | 2.2 | 5.90 | 18/18 |
| 10 | 39 | OA | ND | ND | – | – | – | – | 2.0 | 5.1 | 2.83 | 18/18 |
| 11 | 43 | OA | 46,XY | AZFcgr/gr | – | – | – | – | 2.6 | 4.2 | 3.44 | 25/25 |
| 12 | 45 | OA | 46,XY | AZFcgr/gr | – | – | – | – | 4.7 | 6.1 | 3.46 | 15/10 |
| 13 | 32 | NOA | 46,XY | ND | – | – | – | – | 13.3 | 19.8 | 3.43 | 12/12 |
| 14 | 34 | NOA | 46,XY | No deletion | – | – | – | – | 5.8 | 13.9 | 3.13 | 10/12 |
| 15 | 35 | NOA | 46,XY | No deletion | – | – | – | – | 7.3 | 10.4 | 6.00 | 10/10 |
| 16 | 38 | NOA | 46,XY | No deletion | – | – | – | – | 7.0 | 33.0 | 1.88 | 12/10 |
| 17 | 46 | NOA | 46,XY | AZFcgr/gr | – | – | – | – | 21.4 | 33.8 | 3.14 | 12/12 |

BT: balanced translocation carrier, OA: obstructive azoospermia, NOA: non-obstructive azoospermia.

ND: No deta, -: The examination was not performed because it was not required for their treatment.

The age of patients in control and BT group is at the time of IVF or ICSI. The age of patients in the OA and NOA groups is at the time of TESE.

The complete karyotypes for BT group patients: 45,XY, der(13;14)(q10;q10), 46,XY, t(19;22)(q13.4;q13.2), 46,XY, t(6;16)(q11;p11.2).

(P = 0.32). The mean testicular volume (combined left and right) was 36.6 mL in the OA group and 22.4 mL in the NOA group (P = 0.008) (Table 2).

## Success rate of WGA and karyotyping

Ten sperm per patient were selected and karyotyped to determine the sperm chromosomal conditions. The WGA ratio and the karyotyping ratio for each group are detailed in Table 3. All groups showed no significant difference, with a total karyotyping success rate of 85% (145/170; Table 3). The individual karyotyping success rates did not significantly correlate with the motile sperm count at TESE (R = 0.464, P = 0.177; S2 Dataset). Results of the karyotype analysis demonstrated low noise levels and high accuracy (Fig 3).

## Sperm karyotyping of BT group

Single sperm karyotyping of the BT group was performed to examine the accuracy of the single sperm karyotyping protocol (Fig 2). During gametogenesis in BT carriers, derivative chromosomes (structurally rearranged chromosomes) form a cross with normal chromosomes, and meiosis proceeds through their segregation. Depending on the direction of segregation, this results in alternate segregation, adjacent-1 segregation, adjacent-2 segregation, 3:1 segregation, or 4:0 segregation. The upper portion of Figs 4–6 shows the gamete karyotypes predicted for each segregation pattern based on the karyotypes observed in peripheral blood from BT carriers. The sperm karyotyping results are listed in the lower portion of Figs 4–6 and Table 4. In the BT group, sperm with balanced translocated chromosomes were also detected as haploid, because this analysis method cannot distinguish between balanced chromosomes and normal chromosomes.

**Table 2. Patient history and clinical characteristics: baseline characteristics.**

| | Control | BT | OA | NOA | P-value |
|---|---|---|---|---|---|
| Age, median | 43.5 | 37.0 | 39.0 | 35.0 | 0.19 |
| Semen volume, mean (ml) | 2.8 | 3.4 | – | – | 0.58 |
| Total sperm concentration, mean (million/ml) | 39.7 | 22.6 | – | – | 0.28 |
| Motility, mean (%) | 43.4 | 26.8 | – | – | 0.22 |
| Total motile sperm count, mean (million/ml) | 49.1 | 17.7 | – | – | 0.16 |
| Serum LH, mean (mIU/mL) | – | – | 3.7 | 11.0 | 0.04 |
| Serum FSH, mean (mIU/mL) | – | – | 4.6 | 22.2 | 0.007 |
| Serum Teststerone, mean (mIU/mL) | – | – | 5.0 | 3.5 | 0.32 |
| Total Testis Volume, mean (ml) | – | – | 36.6 | 22.4 | 0.008 |

-: The examination was not performed because it was not required for their treatment.

P-value by Kruskal-Wallis test (for age) and Student's T test (for others).

**Table 3. WGA ratio and Karyotyping ratio of control, BT, OA and NOA group.**

| Group | Number of analyzed sperm | Number of WGA-successful sperm | WGA success rate (%) | Number of karyotyping-successful sperm | Karyotyping success rate (%) |
|---|---|---|---|---|---|
| Control | 40 | 38 | 95% | 36 | 90% |
| BT | 30 | 28 | 93% | 28 | 93% |
| OA | 50 | 42 | 84% | 40 | 80% |
| NOA | 50 | 46 | 92% | 41 | 82% |
| Total | 170 | 154 | 91% | 145 | 85% |

BT: balanced translocation carrier, OA: obstructive azoospermia, NOA: non-obstructive azoospermia.

Analysis of sperm from 45,XY, der(13;14)(q10;q10) carrier (patient #5) revealed one aberrant sperm with 22,X,-14 (Fig 4A). Analysis of sperm from 46,XY, t(19;22)(q13.4;q13.2) carrier (Patient #6) revealed four aberrant sperm:, two sperm with 23,X,der(22)t(19;22)(q13.4;13.2) (Fig 5A) and two sperm with 23,X,der(19)t(19;22)(q13.4;13.2) (Fig 5B). In sperm with 23,X,der(22)t(19;22)(q13.4;13.2), dup(19)(q13.43→qter) was not detected by NGS (Fig 5A). The chromosome size of dup(19)(q13.43→qter) calculated from the reference sequence hg19 was approximately 7.7 Mbp, which is below the detection limit of this PGS kit (17 Mbp copy number events), hence it was presumed undetected. Analysis of sperm from 46,XY, t(6;16)(q11;p11.2) carrier (patient #7) revealed five aberrant sperm: one sperm with 23,X,der(16)t(6;16)(q11;p11.2) (Fig 6A), one sperm with 23,X,der(6)t(6;16)(q11;p11.2) (Fig 6B), one sperm with 23,X,+der(6)t(6;16)(q11;p11.2),-16 (Fig 6C) and two sperm with 23,X,-6,+der(16)t(6;16)(q11;p11.2) (Fig 6D). For 23,X,+der(6)t(6;16)(q11;p11.2),-16 (Fig 6C) and 23,X,-6,+der(16)t(6;16)(q11;p11.2) (Fig 6D), the breakpoint on chromosome 6 detected by NGS deviated from the expected location (Table 4), however, this is also considered an error due to the detection limit of this PGS kit. In all BT carriers, we detected sperm with unbalanced chromosomes derived from translocations.

## Sperm karyotype of control, OA, and NOA group

Single sperm karyotyping of testicular sperm was performed for the control, the OA and the NOA groups. The results are shown in Fig 7 and Table 5. No aberrant sperm were found in the control and OA groups, whereas 7/41 (17%) of the sperm in the NOA group had an aberrant karyotype. Individual aberrant sperm detection rates did not correlate with the motile sperm count at TESE (R=−0.149, P=0.681) (S2 Dataset).

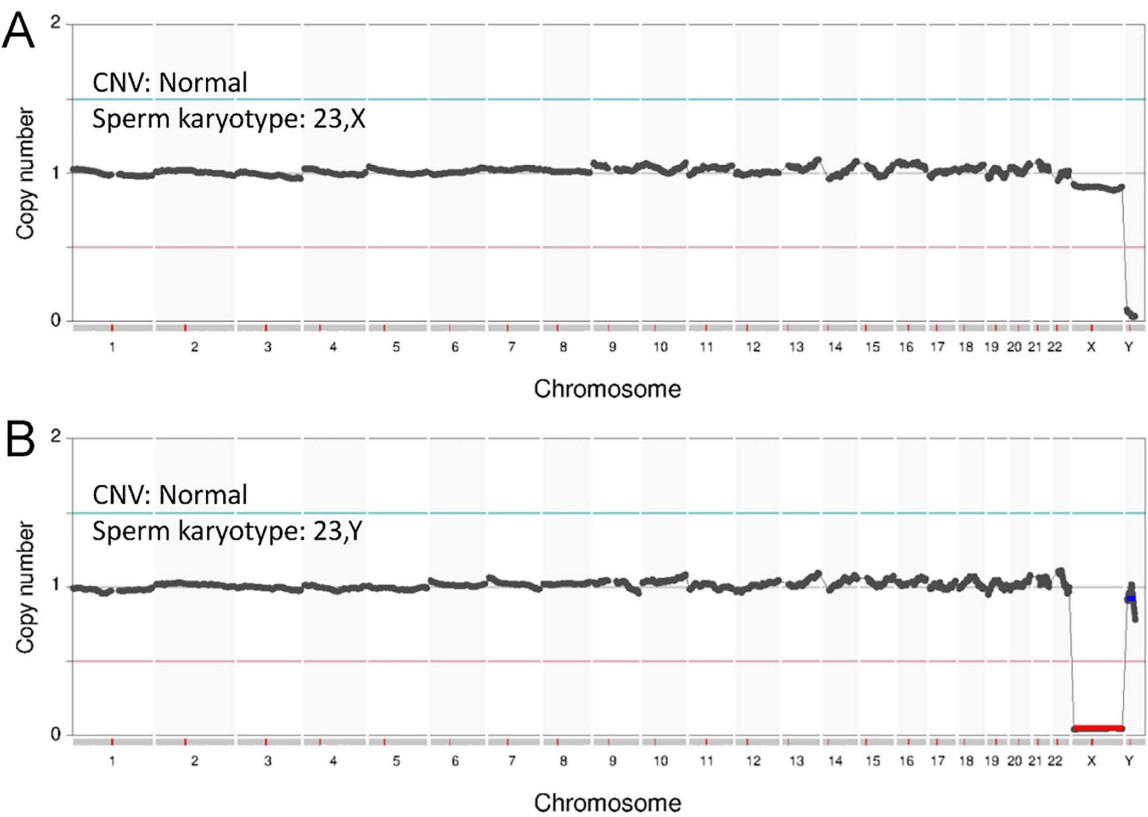

**Fig 3. Example of single sperm sequencing of control group.** Results of the karyotype analysis demonstrated low noise levels and high accuracy. For each NGS image, the copy number variation (CNV) and sperm karyotype are indicated. A. 23,X; the normal haploid karyotype of sperm from patient #1. B. 23,Y; the normal haploid karyotype of sperm from patient #1.

As expected, aberrant sperm were found only in the NOA group (Fig 7). The karyotypes of aberrant sperm in the NOA group are shown in Fig 8 and Table 5. One aberrant sperm was found in each of patients #13 and #16. Five aberrant sperm were found in patient #15. No aberrant sperm were found in patients #14 and #17. Individual differences were observed in the frequency of sperm chromosomal abnormalities in patients with NOA.

Sperm analysis from patient #13 revealed one aberrant sperm with 23,Y, dup(15)(pter→q11.2) (Fig 8A). Sperm analysis from patient #15 revealed five aberrant sperm: two sperm with 24,Y,+7 (Fig 8B), one sperm with 22,Y,-14 (Fig 8C), one sperm with 20,-1,-5 (Fig 8D) and one sperm with 24,Y,+8,del(1)(q42.2→qter) (Fig 8E). Sperm analysis from patient #16 revealed one aberrant sperm with 23,X,del(11)(pterp→11.12) (Fig 8F).

Even with sperm selection for clinical use, the proportion of sperm with chromosomal abnormalities in the NOA group was higher than that in the OA group. Single sperm karyotyping revealed both chromosomal aneuploidies and structural abnormalities in the NOA group.

## Discussion

Conventional FISH methods for testicular sperm karyotyping have three disadvantages. First, only a few selected chromosomal aneuploidies have been identified. Second, structural abnormalities such as translocations have not been detected. Third, sperm morphology was not considered in these analyses, and their results may not reflect the chromosomal conditions of clinically usable sperm. We performed single sperm karyotyping of all chromosomes using NGS. Additionally,

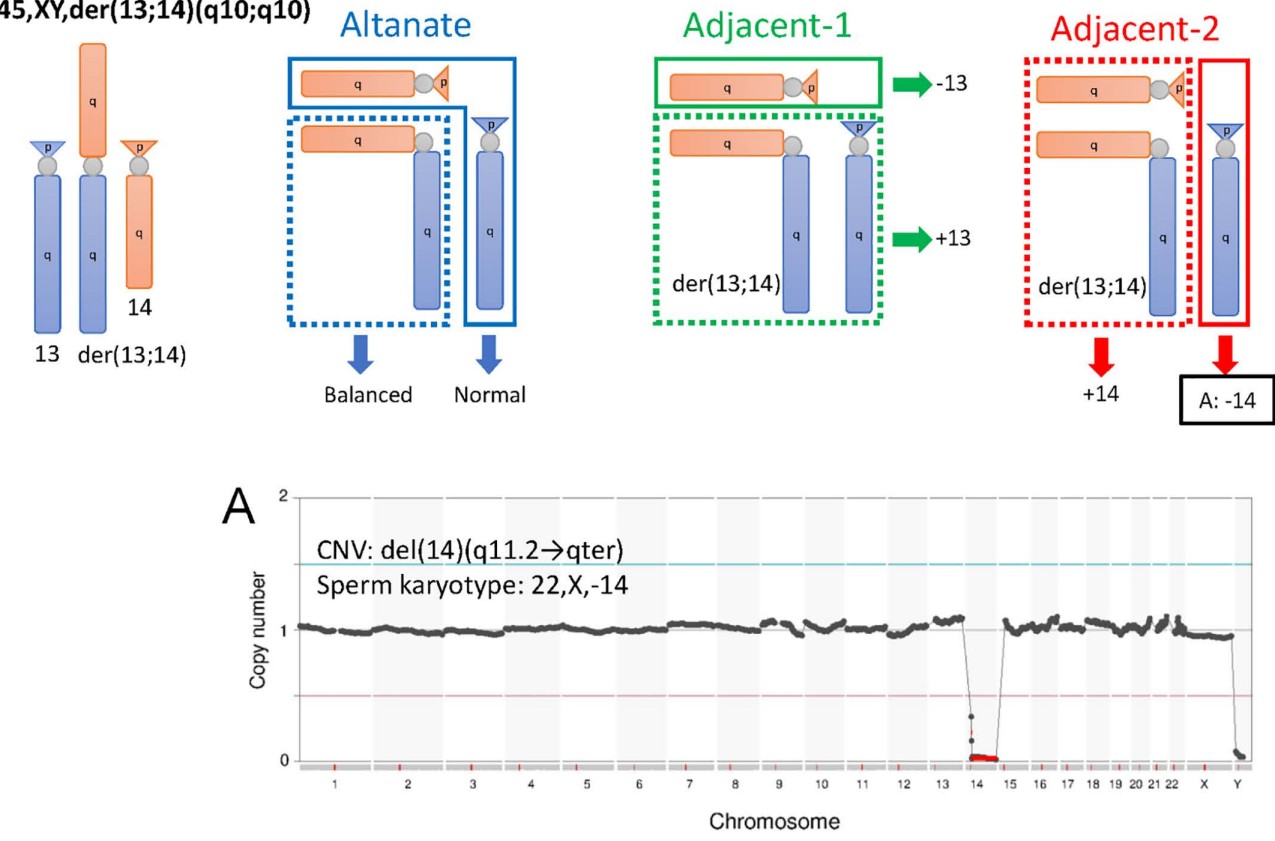

**Fig 4. The gamete karyotypes predicted for each segregation pattern of patient #5 and the sperm karyotyping results.** Gametes formed by alternate segregation, adjacent-1 segregation and adjacent-2 segregation are shown in the upper portion. Segregations 3:1 and 4:0 are omitted as they were not detected by sperm karyotyping. Karyotypes actually detected by sperm karyotyping are outlined in black (A). A correspond to the NGS images in the lower portion. For the NGS image, CNV and sperm karyotype is indicated. The chromosomes derived from balanced translocation carriers were verified. A. 22,X,-14; aberrant karyotype of sperm from patient #5.

we detected known translocations by examining sperm from BT carriers whose karyotypes were previously identified from peripheral blood samples, demonstrating that structural chromosomal abnormalities can be detected. Single-mature sperm with good morphology were selected by micromanipulation and analyzed by NGS to reflect the chromosomal condition of testicular sperm in clinical use.

Although sperm chromosomal abnormalities investigated in previous reports using the conventional FISH method were chromosomes 1, 13, 15, 16, 17, 18, 21, 22, and XY [8], we were able to detect chromosomes 1, 5, 7, 8, 11, 14, 15, and XY abnormalities using NGS of sperm from patients with NOA (Table 5). Our results imply that the rate of chromosomal abnormalities in testicular sperm may be higher than currently recognized if all chromosomes are analyzed using NGS. In addition, sperm selection in this study enabled us to analyze only sperm that had reliably developed into spermatozoa and had good morphology. The karyotyping results may reflect the karyotype of the sperm for clinical use.

In this study, the overall WGA success rate was 91%, and the overall karyotyping success rate was 85%. In a previous report on single sperm karyotyping of ejaculated sperm from translocation carriers, sperm were isolated by fluorescence-activated cell sorting and were analyzed using NGS, yielding a WGA success rate of 78.1% and a karyotyping success rate of 48% [18]. Testicular sperm are few and difficult to isolate, however, isolating them using micromanipulation enabled high-precision isolation, which may have led to an improved karyotyping success rate. The most common

## 46,XY,t(19;22)(q13.4;q13.2)

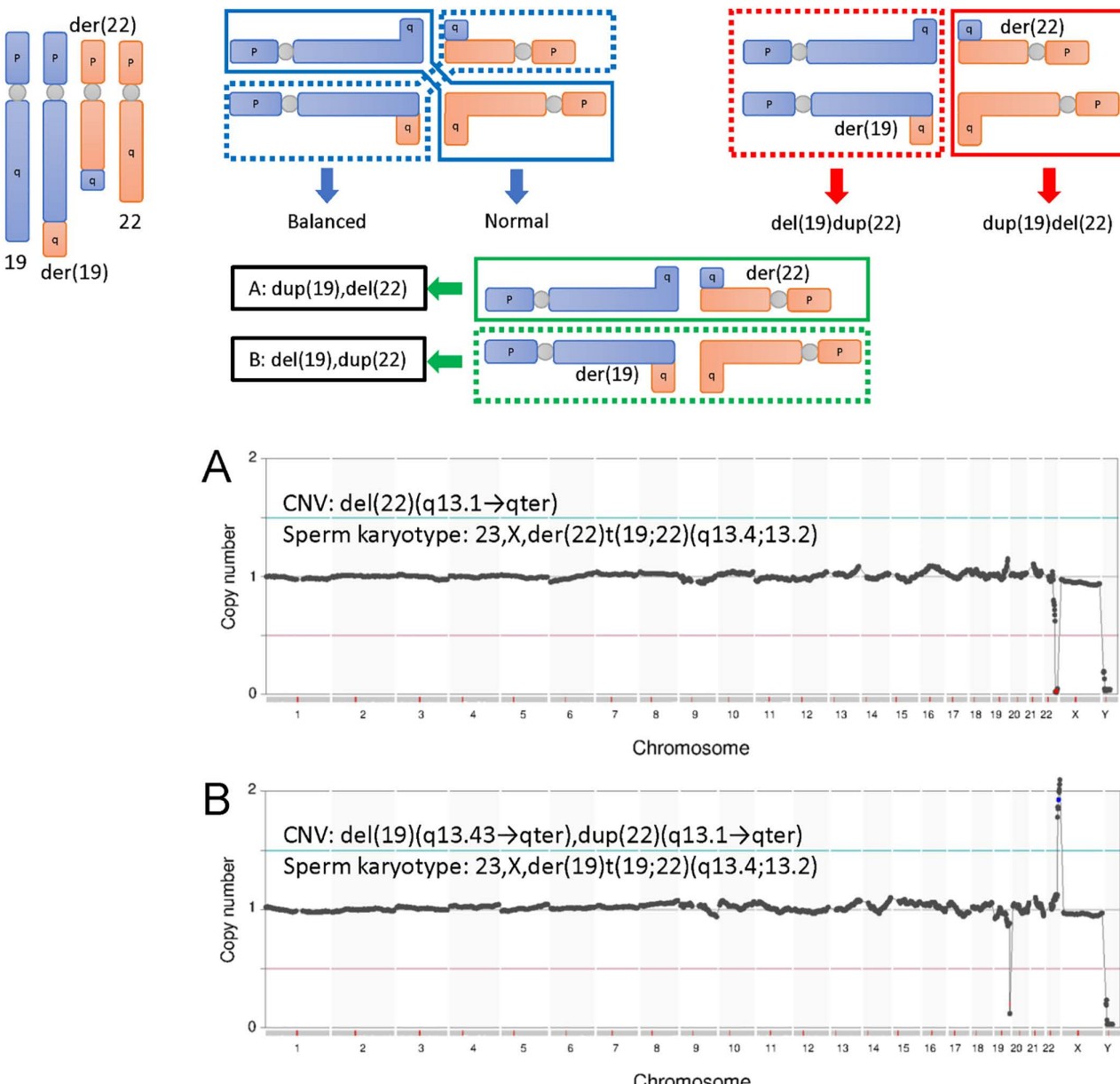

**Fig 5. The gamete karyotypes predicted for each segregation pattern of patient #6 and the sperm karyotyping results.** Gametes formed by alternate segregation, adjacent-1 segregation and adjacent-2 segregation are shown in the upper portion. Karyotypes actually detected by sperm karyotyping are outlined in black **(A-B)**. A-B correspond to the NGS images in the lower portion. For each NGS image, CNV and sperm karyotype are indicated. The chromosomes derived from balanced translocation carriers were verified. A. 23,X,der(19)t(19;22)(q13.4;13.2); aberrant karyotype of sperm from patient #6. B. 23,X,der(22)t(19;22)(q13.4;13.2); aberrant karyotype of the sperm from patient #6.

**Table 4. Karyotype of sperm from BT group.**

| Patient # | Peripheral blood karyotype | Number of aberrant sperm/ Number of karyotyping-successful sperm | Copy number variation | Meiotic segregation pattern | Karyotype of aberrant sperm |
|---|---|---|---|---|---|
| 5 | 45,XY, der(13;14)(q10;q10) | 1/9 | del(14)(q11.2→qter) | adjacent-2 | 22,X,-14 |
| 6 | 46,XY, t(19;22)(q13.4;q13.2) | 4/9 | del(22)(q13.1→qter) | adjacent-1 | 23,X,der(22)t(19;22)(q13.4;13.2) |
| | | | del(22)(q13.1→qter) | adjacent-1 | 23,X,der(22)t(19;22)(q13.4;13.2) |
| | | | del(19)(q13.43→qter),dup(22)(q13.1→qter) | adjacent-1 | 23,X,der(19)t(19;22)(q13.4;13.2) |
| | | | del(19)(q13.43→qter),dup(22)(q13.1→qter) | adjacent-1 | 23,X,der(19)t(19;22)(q13.4;13.2) |
| 7 | 46,XY, t(6;16)(q11;p11.2) | 5/10 | dup(6)(q11.1→qter),del(16)(pter→p11) | adjacent-1 | 23,X,der(16)t(6;16)(q11;p11.2) |
| | | | del(6)(q11.1→qter),dup(16)(pter→p11) | adjacent-1 | 23,X,der(6)t(6;16)(q11;p11.2) |
| | | | dup(6)(pter→p11.1),del(16)(q11.2→qter) | adjacent-2 | 23,X,+der(6)t(6;16)(q11;p11.2),-16 |
| | | | del(6)(pter→p11.1),dup(16)(q11.2→qter) | adjacent-2 | 23,X,-6,+der(16)t(6;16)(q11;p11.2) |
| | | | del(6)(pter→p11.1),dup(16)(q11.2→qter) | adjacent-2 | 23,X,-6,+der(16)t(6;16)(q11;p11.2) |

BT: balanced translocation carrier.

Aberrant sperm: the sperm with chromosome aberration in even one chromosome, del: deletion, dup: duplication.

der: derivative chromosome; structurally rearranged chromosome that is described according to the number and orientation of the chromosome providing the centromere.

**Table 5. Karyotype of sperm from NOA group.**

| Patient # | Peripheral blood karyotype | Number of aberrant sperm/ Number of karyotyping-successful sperm | Copy number variation | Karyotype of aberrant sperm |
|---|---|---|---|---|
| 13 | 46,XY | 1/9 | dup(15)(pter→q11.2) | 23,Y,dup(15)(pter→q11.2) |
| 14 | 46,XY | 0/7 | ND | ND |
| 15 | 46,XY | 5/9 | +7 | 24,Y,+7 |
| | | | +7 | 24,Y,+7 |
| | | | −14 | 22,Y,-14 |
| | | | −1,-5,-X or Y | 20,-1,-5 |
| | | | +8,del(1)(q42.2→qter) | 24,Y,+8,del(1)(q42.2→qter) |
| 16 | 46,XY | 1/8 | del(11)(pter→p11.12) | 23,X,del(11)(pterp→11.12) |
| 17 | 46,XY | 0/8 | ND | ND |

ND: No deta.

Aberrant sperm: the sperm with chromosome aberration in even one chromosome, del: deletion, dup: duplication.

46,XY,t(6;16)(q11;p11.2)

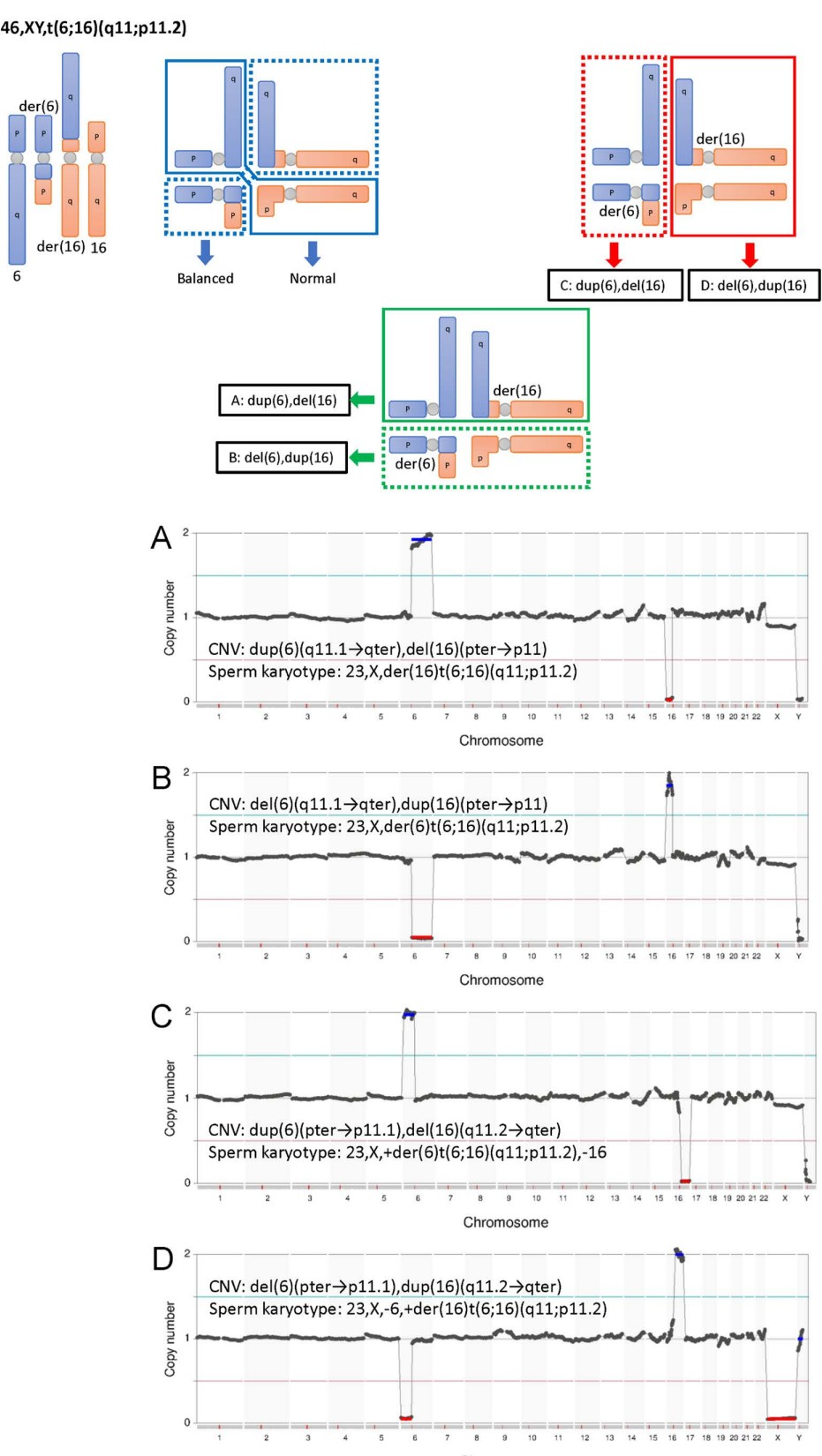

**Fig 6. The gamete karyotypes predicted for each segregation pattern of patient #7 and the sperm karyotyping results.** Gametes formed by alternate segregation, adjacent-1 segregation and adjacent-2 segregation are shown in the upper portion. Karyotypes actually detected by sperm karyotyping are outlined in black **(A-D)**. A-D correspond to the NGS images in the lower portion. For each NGS image, CNV and sperm karyotype are indicated. The chromosomes derived from balanced translocation carriers were verified. A. 23,X,der(6)t(6;16)(q11;p11.2); aberrant karyotype of sperm from patient #7. B. 23,X,-6,+der(16)t(6;16)(q11;p11.2); aberrant karyotype of sperm from patient #7. C. 23,X,+der(6)t(6;16)(q11;p11.2),-16; aberrant karyotype of sperm from patient #7. D. 23,X,der(16)t(6;16)(q11;p11.2); aberrant karyotype of sperm from patient #7.

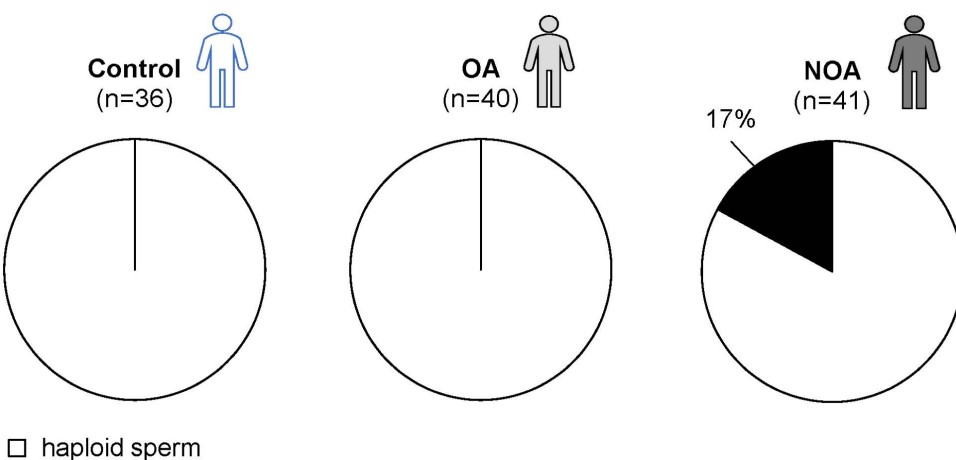

☐ haploid sperm

■ aberrant sperm

**Fig 7. Rate of aberrant sperm in each group.** Aberrant sperm were found only in the NOA group.

reason for unsuccessful karyotyping in this study was amplification failure (S1 Dataset). This may be due to natural DNA degradation or fragmentation during the cryopreservation and the preparation of single sperm DNA.

Karyotyping of the BT group, which was conducted as the quality control for karyotyping, detected unbalanced chromosomes resulting from the carrier translocation in all males. In a patient with a Robertsonian translocation 45,XY, der(13;14)(q10;q10), only one sperm carrying an unbalanced chromosome was detected. This finding is consistent with a previous report; approximately 80–90% of sperm from Robertsonian translocation carriers have balanced chromosomes [25]. The segregation patterns of unbalanced chromosomes detected in this study were adjacent-1 and adjacent-2 segregation patterns only; 3:1 and 4:0 segregation patterns were not detected (Table 4, Fig 4). This bias in segregation patterns may have been attributable to the small sample size, given that a previous karyotyping analysis of 32 sperm from a balanced translocation carrier reported a 3:1 segregation pattern [18].

In the control group, no aberrant sperm were observed. This result is expected, as sperm from men with normal semen parameters are known to have a low proportion of aberrant sperm [6]. Similarly, in the OA group, no sperm with chromosomal abnormalities were detected. This is thought to be because OA is caused by obstructive azoospermia, which means that spermatogenesis is preserved, and because microscopic selection of morphologically normal mature sperm was applied. In the NOA group, in our cohort, chromosomal abnormalities were observed even with microscopic selection. In patients with NOA, meiotic abnormalities are often observed in their testis [26], and apoptosis due to meiotic abnormalities is considered to be one of the causes of azoospermia [27]. Therefore, sperm chromosomal abnormalities may have only been observed in patients with NOA.

In the NOA group, the occurrence of chromosomal abnormalities varies between individuals. The mechanisms that selectively induce apoptosis in sperm with chromosomal abnormalities due to meiotic abnormalities are called meiotic checkpoints. The

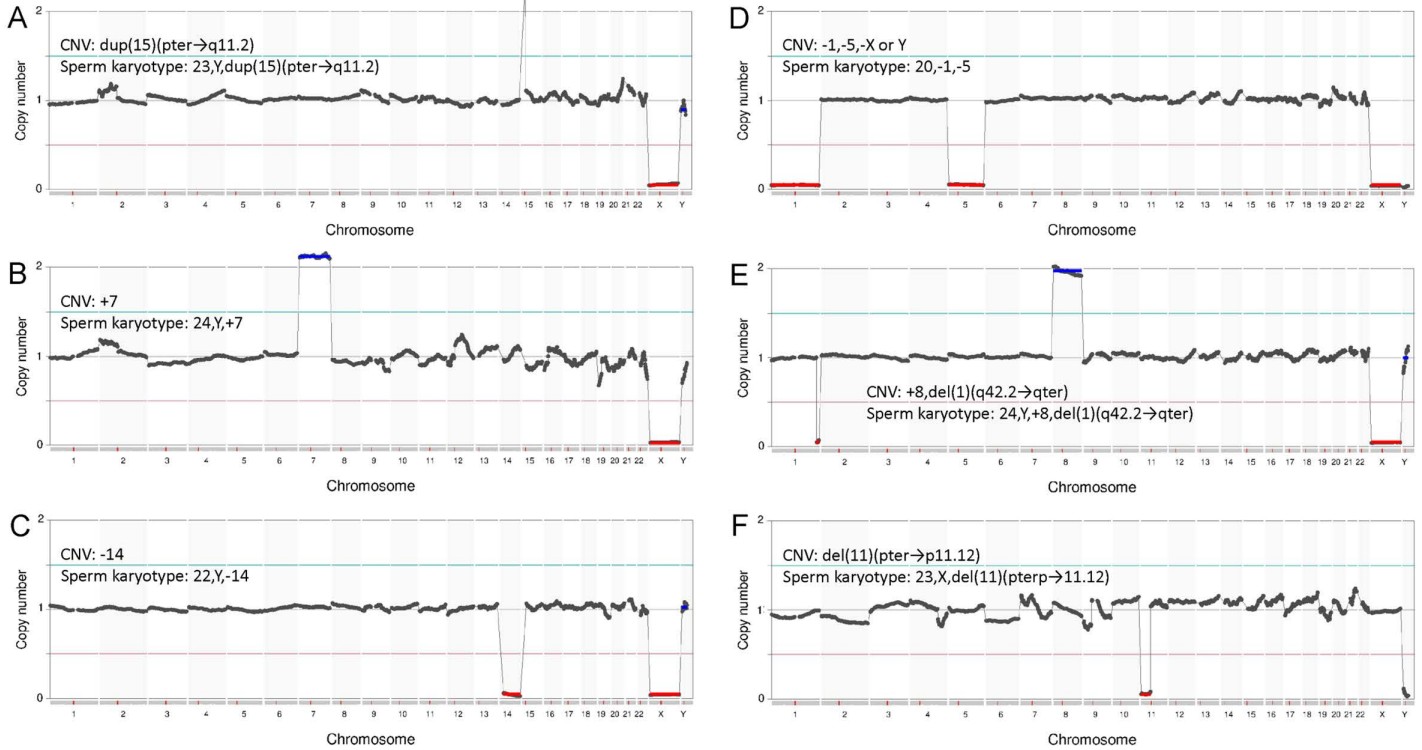

**Fig 8. Aberrant karyotype of single sperm sequencing of NOA group.** Single sperm sequencing of NOA group confirmed both chromosomal aneuploidy and structural chromosomal abnormalities. For each NGS image, CNV and sperm karyotype are indicated. A. 23,Y, dup(15)(pter→q11.2); aberrant karyotype of sperm from patient #13. B. 24,Y,+7; aberrant karyotype of sperm from patient #15. C. 22,Y,-14; aberrant karyotype of sperm from patient #15. D. 20,-1,-5; aberrant karyotype of sperm from patient #15. E. 24,Y,+8,del(1)(q42.2→qter): Aberrant karyotype of sperm from patient #15. F. 23,X,del(11)(pterp→11.12): Aberrant karyotype of sperm from patient #16.

strictness of these meiotic checkpoints varies between individuals [9,26]. Some individuals have strict checkpoints that eliminate most aberrant sperm, whereas others have difficulty in eliminating aberrant sperm. Among studies analyzing the results of PGT-A after TESE-ICSI, some have reported an increase in chromosomal abnormalities in embryos [13], whereas others have not reported any significant difference [28]. These discrepancies may be attributable to the fact that the frequency of chromosomal abnormalities in testicular sperm used for TESE-ICSI varies from person to person, which may affect the results of PGT-A.

Severe male infertility is an indication for PGT-A, however, PGT-A carries potential risks such as embryo invasiveness, uncertain long-term prognosis of the offspring. Therefore, PGT-A should be performed only for patients who truly require it. Sperm FISH analysis has been suggested to potentially aid in identifying patients requiring PGT-A [10]. However, single sperm sequencing provides more information than FISH because it analyzes all chromosomes, and it may also be useful in considering the necessity of PGT-A. In our cohort, the occurrence of chromosomal abnormalities varies between individuals in the NOA group. In Patient #15 of the NOA group, even after microscopic selection, multiple sperm had chromosomal abnormalities. In such patients, it is possible that sperm with chromosomal abnormalities are injected into oocytes during TESI-ICSI, suggesting a high risk of chromosomal abnormalities in embryos due to the sperm. By performing single sperm sequencing before proceeding to ICSI, we can examine the sperm chromosomal potential of patients with NOA. Since the absence of chromosomal abnormalities in the oocytes is also necessary for pregnancy, single sperm sequencing cannot serve as an alternative to PGT-A. However, when considering whether PGT-A is appropriate, single sperm sequencing may provide complementary information.

This study has some limitations. First, only ten sperm per patient were analyzed, and results may not fully represent per-patient variability. Second, micromanipulation is operator-dependent, which could affect reproducibility. Third, this protocol had a detection limit, and structural abnormalities lower than this limit could not be detected. Fourth, whole-genome WGA introduced potential artifacts that could not be completely excluded. Finally, because sperm were lysed for NGS, they could not be used for insemination. Therefore, the karyotyping results cannot directly connect embryo outcomes. Larger per-patient studies are warranted to validate these findings.

Previously, testicular sperm karyotyping only provided information on abnormalities in specific chromosomes. By analyzing all chromosomes of testicular sperm using NGS, we could detect chromosomal aneuploidy and structural abnormalities that had not been previously examined. By examining mature testicular sperm with good morphology, we were able to obtain chromosomal information reflecting the karyotypes of the sperm for clinical use.

Although single sperm sequencing is not a flawless method, when considering whether PGT-A is appropriate, it may provide additional insights without invasiveness to embryos.

## Conclusions

In this study, we detected chromosomal aneuploidy and structural abnormalities by analyzing all chromosomes of testicular sperm using NGS. In our cohort, chromosomal abnormalities were observed only in patients with NOA and the frequency of chromosomal abnormalities varied among patients. In NOA, sperm chromosomal abnormalities may contribute to the lower fertilization and pregnancy rates in TESE-ICSI. Although validation using larger per-patient datasets is necessary, incorporating this assay may help identify patients with NOA most likely to benefit from PGT-A.

## Supporting information

**S1 Dataset. Karyotype analysis results for each sperm.** Karyotype analysis results for each sperm of control, BT, OA and NOA groups. WGA cut off <5ng/µl.
(PDF)

**S2 Dataset. Individual karyotyping success rates and individual aberrant sperm detection rates.** Individual karyotyping success rates (number of karyotyping-successful sperm/ number of analyzed sperm) did not significantly correlate with motile sperm count at TESE ($R = 0.464$, $P = 0.177$). Individual aberrant sperm detection rates (number of aberrant sperm/ number of karyotyping-successful sperm) did not correlate with motile sperm count at TESE ($R = -0.149$, $P = 0.681$).
(PDF)

## Acknowledgments

We thank the clinicians, embryologists, and nursing staff at the Center for Reproductive Medicine, Yokohama City University Medical Center, Japan. We are grateful to Takahiro Yamaji and Yuka Kinoshita for special advice on our manuscript, and Syunsuke Miyai for validation of experimental protocol. We would like to thank Editage (www.editage.com) for English language editing.

## Author contributions

**Conceptualization:** Sumiko Sueyoshi, Hiroe Ueno, Tomonari Hayama.

**Data curation:** Sumiko Sueyoshi.

**Formal analysis:** Sumiko Sueyoshi, Akifumi Ijuin, Akira Yanagihara.

**Investigation:** Sumiko Sueyoshi, Akifumi Ijuin.

**Methodology:** Sumiko Sueyoshi.

**Project administration:** Sumiko Sueyoshi.

**Resources:** Ai Miyakoshi, Haru Hamada, Misaki Toda, Naoki Tsuchiya, Miki Tanoshima, Mayuko Kurumizaka, Marina Saito, Yuki Oike, Kazumi Takeshima, Teppei Takeshima, Shinnosuke Kuroda, Yasushi Yumura, Shin Saito, Ryoko Asano, Taichi Mizushima, Hideya Sakakibara.

**Supervision:** Hiroe Ueno, Miki Tanoshima, Teppei Takeshima, Etsuko Miyagi, Hiroki Kurahashi, Akira Yanagihara, Mariko Murase, Tomonari Hayama.

**Validation:** Sumiko Sueyoshi, Akifumi Ijuin, Hiroe Ueno, Akira Yanagihara, Tomonari Hayama.

**Visualization:** Akira Yanagihara.

**Writing – original draft:** Sumiko Sueyoshi.

**Writing – review & editing:** Sumiko Sueyoshi, Akifumi Ijuin, Naoki Tsuchiya, Mariko Murase, Tomonari Hayama.

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
