## [Decision Letter · Decision Letter 0]

3 Sep 2025

Dear Dr. Hayama ,

We look forward to receiving your revised manuscript.

Kind regards,

Joël R Drevet, Ph.D.

Academic Editor

PLOS ONE

Journal Requirements:

1.Please ensure that your manuscript meets PLOS ONE's style requirements, including those for file naming. The PLOS ONE style templates can be found at https://journals.plos.org/plosone/s/file?id=wjVg/PLOSOne_formatting_sample_main_body.pdf and https://journals.plos.org/plosone/s/file?id=ba62/PLOSOne_formatting_sample_title_authors_affiliations.pdf

Additional Editor Comments:

The submission was evaluated by a cytogenetics expert who provided a detailed assessment, which you will find below. Given the significant weaknesses in the presentation of the data and their excessive interpretation in some cases, a “major revision” is recommended, while taking into account that the technology developed may be of some interest. Please carefully follow the reviewer's comments. Some sections could be condensed to avoid unnecessary repetition. In addition, the discussion should focus on your own data.

Reviewers' comments:

Reviewer's Responses to Questions

**Comments to the Author**

1. Is the manuscript technically sound, and do the data support the conclusions?

Reviewer #1: Yes

2. Has the statistical analysis been performed appropriately and rigorously?

Reviewer #1: Yes

3. Have the authors made all data underlying the findings in their manuscript fully available?

Reviewer #1: No

4. Is the manuscript presented in an intelligible fashion and written in standard English?

Reviewer #1: No

Reviewer #1: I had the opportunity to review the paper entitled “Single sperm karyotyping of testicular sperm of non-obstructive and obstructive azoospermia using next generation sequencing” by Tomonari Hayama et al. The study aims to evaluate the incidence of chromosomal abnormalities in sperm from patients with non-obstructive azoospermia, considering previous studies that reported an increased incidence of such abnormalities. Although the results are not entirely novel, the methodology proposed here—based on approaches commonly used for PGT-A analysis—is of interest. However, the paper requires substantial revision before it can be considered for publication in PLOS ONE, particularly the discussion section, as explained below. In addition, there are numerous typographical errors, inappropriate sentences, and cytogenetic inaccuracies that must be corrected.

Major remarks:

1- The discussion section, in its current form, is not suitable and needs to be revised and considerably reduced.

- the authors should provide a clear explanation of the reported 20% failure rate, and discuss the protocol in more detail, particularly its feasibility and reproducibility for other teams attempting to replicate this strategy.

- the results should be discussed separately for each group in light of previous studies, with particular attention to the NOA group, where high variability has been observed. This discussion should also be connected to the high aneuploidy rate reported in oocytes.

- it is unnecessary to repeat large portions of the introduction at the beginning of the discussion. This section should instead focus on the interpretation of the results.

- please clarify why it could be relevant to propose PGT-A for couples similar to patient 15. In this specific case, was a sperm FISH analysis performed for chromosome 7?

- Finally, the discussion should focus more closely on the study results, avoiding excessive speculation. In several places, sentences extend beyond the scope of the findings and should be revised.

2- There are several cytogenetic mistakes and misinterpretations that significantly compromise the accuracy of the results. As a cytogeneticist, it is evident that the manuscript has not been critically reviewed by a specialist in this field. Below, I summarize the most important issues; however, I strongly recommend that the authors consult with an expert in cytogenetics to ensure accuracy and consistency throughout the paper.

Line 89: The authors refer to “an antibody,” but this is incorrect. It should be described as a specific DNA probe labeled with a fluorochrome.

Line 231: The karyotypes are not reported according to ISCN nomenclature (e.g., “46XY” should be “46,XY”). In addition, the complete karyotype formula should be provided for the BT group, particularly for patients with reciprocal translocations, as this is essential for proper interpretation of the results.

When describing the absence or duplication of a whole chromosome, the appropriate terms are nullisomy and disomy, respectively, and should be used consistently throughout the manuscript.

The section Sperm karyotyping of the BT group requires a thorough revision according to both translocation segregation patterns and ISCN nomenclature. For example, in patient 6, sample 3, the karyotype should be written as 23,X,+der(19)t(19;22),-19. This clarification is necessary for accurate interpretation. For the same patient, the reported karyotype 23,X,del(22) is questionable. It may in fact correspond to 23,X,+der(22)t(19;22),-22. The authors should carefully review this point.

The manuscript contains numerous typographical errors and poorly constructed sentences. A thorough language revision is strongly recommended.

Line 369: Please clarify the mention of “chr 23.” This terminology is incorrect and should be revised.

Line 387: The phrase “zygote stage of meiosis” is inappropriate and scientifically incorrect. Please rephrase this sentence to accurately reflect the intended meaning.

Best,

**Do you want your identity to be public for this peer review?** For information about this choice, including consent withdrawal, please see our Privacy Policy

Reviewer #1: No

---

## [Author Response · Author response to Decision Letter 1]

23 Oct 2025

PONE-D-25-35308

Single sperm karyotyping of testicular sperm of non-obstructive and obstructive azoospermia using next generation sequencing

PLOS ONE

Dear Dr. Hayama ,

Thank you for submitting your manuscript to PLOS ONE. After careful consideration, we feel that it has merit but does not fully meet PLOS ONE’s publication criteria as it currently stands. Therefore, we invite you to submit a deeply revised version of the manuscript that addresses the points raised during the review process (see below).

We look forward to receiving your revised manuscript.

Kind regards,

Joël R Drevet, Ph.D.

Academic Editor

PLOS ONE

Journal Requirements:

1.Please ensure that your manuscript meets PLOS ONE's style requirements, including those for file naming. The PLOS ONE style templates can be found at https://journals.plos.org/plosone/s/file?id=wjVg/PLOSOne_formatting_sample_main_body.pdf and https://journals.plos.org/plosone/s/file?id=ba62/PLOSOne_formatting_sample_title_authors_affiliations.pdf

Additional Editor Comments:

The submission was evaluated by a cytogenetics expert who provided a detailed assessment, which you will find below. Given the significant weaknesses in the presentation of the data and their excessive interpretation in some cases, a “major revision” is recommended, while taking into account that the technology developed may be of some interest. Please carefully follow the reviewer's comments. Some sections could be condensed to avoid unnecessary repetition. In addition, the discussion should focus on your own data.

Reviewers' comments:

Reviewer's Responses to Questions

Comments to the Author

1. Is the manuscript technically sound, and do the data support the conclusions?

Reviewer #1: Yes

2. Has the statistical analysis been performed appropriately and rigorously?

Reviewer #1: Yes

3. Have the authors made all data underlying the findings in their manuscript fully available?

Reviewer #1: No

4. Is the manuscript presented in an intelligible fashion and written in standard English?

Reviewer #1: No

5. Review Comments to the Author

Reviewer #1: I had the opportunity to review the paper entitled “Single sperm karyotyping of testicular sperm of non-obstructive and obstructive azoospermia using next generation sequencing” by Tomonari Hayama et al. The study aims to evaluate the incidence of chromosomal abnormalities in sperm from patients with non-obstructive azoospermia, considering previous studies that reported an increased incidence of such abnormalities. Although the results are not entirely novel, the methodology proposed here—based on approaches commonly used for PGT-A analysis—is of interest. However, the paper requires substantial revision before it can be considered for publication in PLOS ONE, particularly the discussion section, as explained below. In addition, there are numerous typographical errors, inappropriate sentences, and cytogenetic inaccuracies that must be corrected.

Major remarks:

1- The discussion section, in its current form, is not suitable and needs to be revised and considerably reduced.

- the authors should provide a clear explanation of the reported 20% failure rate, and discuss the protocol in more detail, particularly its feasibility and reproducibility for other teams attempting to replicate this strategy.

- the results should be discussed separately for each group in light of previous studies, with particular attention to the NOA group, where high variability has been observed. This discussion should also be connected to the high aneuploidy rate reported in oocytes.

- it is unnecessary to repeat large portions of the introduction at the beginning of the discussion. This section should instead focus on the interpretation of the results.

- please clarify why it could be relevant to propose PGT-A for couples similar to patient 15. In this specific case, was a sperm FISH analysis performed for chromosome 7?

- Finally, the discussion should focus more closely on the study results, avoiding excessive speculation. In several places, sentences extend beyond the scope of the findings and should be revised.

2- There are several cytogenetic mistakes and misinterpretations that significantly compromise the accuracy of the results. As a cytogeneticist, it is evident that the manuscript has not been critically reviewed by a specialist in this field. Below, I summarize the most important issues; however, I strongly recommend that the authors consult with an expert in cytogenetics to ensure accuracy and consistency throughout the paper.

Line 89: The authors refer to “an antibody,” but this is incorrect. It should be described as a specific DNA probe labeled with a fluorochrome.

Line 231: The karyotypes are not reported according to ISCN nomenclature (e.g., “46XY” should be “46,XY”). In addition, the complete karyotype formula should be provided for the BT group, particularly for patients with reciprocal translocations, as this is essential for proper interpretation of the results.

When describing the absence or duplication of a whole chromosome, the appropriate terms are nullisomy and disomy, respectively, and should be used consistently throughout the manuscript.

The section Sperm karyotyping of the BT group requires a thorough revision according to both translocation segregation patterns and ISCN nomenclature. For example, in patient 6, sample 3, the karyotype should be written as 23,X,+der(19)t(19;22),-19. This clarification is necessary for accurate interpretation. For the same patient, the reported karyotype 23,X,del(22) is questionable. It may in fact correspond to 23,X,+der(22)t(19;22),-22. The authors should carefully review this point.

The manuscript contains numerous typographical errors and poorly constructed sentences. A thorough language revision is strongly recommended.

Line 369: Please clarify the mention of “chr 23.” This terminology is incorrect and should be revised.

Line 387: The phrase “zygote stage of meiosis” is inappropriate and scientifically incorrect. Please rephrase this sentence to accurately reflect the intended meaning.

Best,

6. PLOS authors have the option to publish the peer review history of their article (what does this mean?). If published, this will include your full peer review and any attached files.

Do you want your identity to be public for this peer review? For information about this choice, including consent withdrawal, please see our Privacy Policy.

Reviewer #1: No

---

## [Decision Letter · Decision Letter 1]

16 Nov 2025

Dear Dr. Hayama,

Thank you for submitting your manuscript to PLOS ONE. After careful consideration, we feel that it has merit but does not fully meet PLOS ONE’s publication criteria as it currently stands. Therefore, we invite you to submit a revised version of the manuscript that addresses the points raised during the review process.

We look forward to receiving your revised manuscript.

Kind regards,

Prof. Joël R Drevet, Ph.D.

Academic Editor

PLOS ONE

Journal Requirements:

**Additional Editor Comments:**

After this first round of revisions, the manuscript has been significantly improved. However, there are still a few additional points that could be addressed to make it even better.

I am therefore returning it to you so that you make a few more minor changes. Please note that this will be the final revision phase offered by the editorial office, so try to cover all the points raised by the reviewer.

Reviewers' comments:

Reviewer's Responses to Questions

**Comments to the Author**

Reviewer #1: All comments have been addressed

2. Is the manuscript technically sound, and do the data support the conclusions?

Reviewer #1: Yes

3. Has the statistical analysis been performed appropriately and rigorously?

Reviewer #1: Yes

4. Have the authors made all data underlying the findings in their manuscript fully available?

Reviewer #1: Yes

5. Is the manuscript presented in an intelligible fashion and written in standard English?

Reviewer #1: No

Reviewer #1: I have had the opportunity to review for the second time the manuscript entitled “Single sperm karyotyping of testicular sperm in non-obstructive and obstructive azoospermia using next generation sequencing” submitted to “Plos One”. The paper has been clearly improved and seems to be suitable for publication after few improvements, especially concerning the redundancy.

One point is worth discussing. You clearly confirm a high rate of chromosomal abnormalities in the sperm of patients with NOA. However, before proposing single sperm sequencing prior to PGT-A, you need to consider several points:

- The cost: on average, 5 to 7 blastocysts can be expected per ICSI. Compared to the analysis of 10 spermatozoa, PGT-A therefore seems less expensive.

- The effectiveness of PGT-A in detecting oocyte aneuploidy, which is impossible to achieve with a sperm analysis.

- The risk associated with biopsy and embryo freezing/thawing. This risk appears to be very low today.

Therefore, in my opinion, you are proposing a very interesting new technique, but one that is difficult to implement in medical practice because it is less relevant than PGT-A analysis and non-invasive PGT-A using culture media.

Regarding the structure of the text, here are a few comments:

The introduction could be shortened, and some elements could be deleted.

There are errors, redundancies, and repetitions; see some examples according to lines number in the highlighted version.

- Lines 198-200 and 203-204: repetitions

- Lines 231-234: number of patients already mentioned previously

- paragraphs 246-254: data from the table, and please note that it is not necessary to point out the difference between OA and NOA. The OA and NOA diagnosis has already been made previously, so please avoid lines 246-248.

- paragraphs 257-267: as the details for each group in Table 3, please summarize as follows: “no difference in all groups with a total success rate of...”.

- lines 278-282, in the introduction

- line 347, , Not “surprisingly”, “as expected”

- lines 388-391 and 397-398 unnecessary

Your best

**Do you want your identity to be public for this peer review?** For information about this choice, including consent withdrawal, please see our Privacy Policy

Reviewer #1: No

---

## [Author Response · Author response to Decision Letter 2]

18 Nov 2025

PONE-D-25-35308R1

Single sperm karyotyping of testicular sperm in non-obstructive and obstructive azoospermia using next generation sequencing

PLOS ONE

Dear Dr. Hayama,

Thank you for submitting your manuscript to PLOS ONE. After careful consideration, we feel that it has merit but does not fully meet PLOS ONE’s publication criteria as it currently stands. Therefore, we invite you to submit a revised version of the manuscript that addresses the points raised during the review process.

We look forward to receiving your revised manuscript.

Kind regards,

Prof. Joël R Drevet, Ph.D.

Academic Editor

PLOS ONE

Journal Requirements:

Additional Editor Comments:

After this first round of revisions, the manuscript has been significantly improved. However, there are still a few additional points that could be addressed to make it even better.

I am therefore returning it to you so that you make a few more minor changes. Please note that this will be the final revision phase offered by the editorial office, so try to cover all the points raised by the reviewer.

Reviewers' comments:

Reviewer's Responses to Questions

Comments to the Author

1. If the authors have adequately addressed your comments raised in a previous round of review and you feel that this manuscript is now acceptable for publication, you may indicate that here to bypass the “Comments to the Author” section, enter your conflict of interest statement in the “Confidential to Editor” section, and submit your "Accept" recommendation.

Reviewer #1: All comments have been addressed

2. Is the manuscript technically sound, and do the data support the conclusions?

Reviewer #1: Yes

3. Has the statistical analysis been performed appropriately and rigorously?

Reviewer #1: Yes

4. Have the authors made all data underlying the findings in their manuscript fully available?

Reviewer #1: Yes

5. Is the manuscript presented in an intelligible fashion and written in standard English?

Reviewer #1: No

6. Review Comments to the Author

Reviewer #1: I have had the opportunity to review for the second time the manuscript entitled “Single sperm karyotyping of testicular sperm in non-obstructive and obstructive azoospermia using next generation sequencing” submitted to “Plos One”. The paper has been clearly improved and seems to be suitable for publication after few improvements, especially concerning the redundancy.

One point is worth discussing. You clearly confirm a high rate of chromosomal abnormalities in the sperm of patients with NOA. However, before proposing single sperm sequencing prior to PGT-A, you need to consider several points:

- The cost: on average, 5 to 7 blastocysts can be expected per ICSI. Compared to the analysis of 10 spermatozoa, PGT-A therefore seems less expensive.

- The effectiveness of PGT-A in detecting oocyte aneuploidy, which is impossible to achieve with a sperm analysis.

- The risk associated with biopsy and embryo freezing/thawing. This risk appears to be very low today.

Therefore, in my opinion, you are proposing a very interesting new technique, but one that is difficult to implement in medical practice because it is less relevant than PGT-A analysis and non-invasive PGT-A using culture media.

Regarding the structure of the text, here are a few comments:

The introduction could be shortened, and some elements could be deleted.

There are errors, redundancies, and repetitions; see some examples according to lines number in the highlighted version.

- Lines 198-200 and 203-204: repetitions

- Lines 231-234: number of patients already mentioned previously

- paragraphs 246-254: data from the table, and please note that it is not necessary to point out the difference between OA and NOA. The OA and NOA diagnosis has already been made previously, so please avoid lines 246-248.

- paragraphs 257-267: as the details for each group in Table 3, please summarize as follows: “no difference in all groups with a total success rate of...”.

- lines 278-282, in the introduction

- line 347, , Not “surprisingly”, “as expected”

- lines 388-391 and 397-398 unnecessary

Your best

7. PLOS authors have the option to publish the peer review history of their article (what does this mean?). If published, this will include your full peer review and any attached files.

Do you want your identity to be public for this peer review? For information about this choice, including consent withdrawal, please see our Privacy Policy.

Reviewer #1: No

---

## [Decision Letter · Decision Letter 2]

19 Nov 2025

Single sperm karyotyping of testicular sperm in non-obstructive and obstructive azoospermia using next generation sequencing

PONE-D-25-35308R2

Dear Dr. Hayama,

We’re pleased to inform you that your manuscript has been judged scientifically suitable for publication and will be formally accepted for publication once it meets all outstanding technical requirements.

Kind regards,

Prof. Joël R Drevet, Ph.D.

Academic Editor

PLOS ONE

Additional Editor Comments (optional):

Thanks for having followed the suggestions and remarks.

Reviewers' comments:

Reviewer's Responses to Questions

**Comments to the Author**

Reviewer #1: All comments have been addressed

2. Is the manuscript technically sound, and do the data support the conclusions?

Reviewer #1: Yes

3. Has the statistical analysis been performed appropriately and rigorously?

Reviewer #1: Yes

4. Have the authors made all data underlying the findings in their manuscript fully available?

Reviewer #1: Yes

5. Is the manuscript presented in an intelligible fashion and written in standard English?

Reviewer #1: Yes

Reviewer #1: I have had the opportunity to review for the thrid time the manuscript entitled “Single sperm karyotyping of testicular sperm in non-obstructive and obstructive azoospermia using next generation sequencing” submitted to “Plos One”. The paper has been clearly improved and seems to be suitable for publication as it. Authors answered to all my queries.

Your best

**Do you want your identity to be public for this peer review?** For information about this choice, including consent withdrawal, please see our Privacy Policy

Reviewer #1: No

---

## [Editor Report · Acceptance letter]

PONE-D-25-35308R2

PLOS ONE

Dear Dr. Hayama,

I'm pleased to inform you that your manuscript has been deemed suitable for publication in PLOS ONE. Congratulations! Your manuscript is now being handed over to our production team.

Kind regards,

on behalf of

Prof. Joël R Drevet

Academic Editor

PLOS ONE